# Remote and Autonomous Measurements of Precipitation for the Northwest Ross Ice Shelf, Antarctica

Mark W. Seefeldt[1], Taydra M. Low[2], Scott D. Landolt[3], Thomas H. Nylen[4]

[1]National Snow Ice and Data Center, University of Colorado Boulder, Boulder, CO, 80303, USA
[2]Department of Atmospheric and Oceanic Sciences, University of Colorado Boulder, Boulder, CO, 80303, USA
[3]Research Applications Laboratory, National Center for Atmospheric Research, Boulder, CO, 80301, USA
[4]Technical University of Denmark, Copenhagen, 2800, Denmark

*Correspondence to*: Mark W. Seefeldt (mark.seefeldt@colorado.edu)

**Abstract.** The Antarctic Precipitation System project deployed and maintained four sites across the northwest Ross Ice Shelf
in Antarctica from November 2017 to November 2019. The goals for the project included the collection of in situ observations of precipitation in Antarctica spanning a duration of two years, an improvement in the understanding of precipitation events across the Ross Ice Shelf, and the ability to validate precipitation data from atmospheric numerical models. At each of the four sites the precipitation was measured with an OTT Pluvio$^2$ precipitation gauge. Additionally, snow accumulation at the site was measured with a sonic ranging sensor and using GPS-Interferometry Reflectivity.
Supplemental observations of temperature, wind speed, particle count, particle size and speed, and images and video from a camera, were collected to provide context to the precipitation measurements. The collected dataset represents some of the first year-round observations of precipitation in Antarctica at remote locations using an autonomous measurement system. The acquired observations have been quality controlled, post-processed, and are available for retrieval through the United States Antarctic Program Data Center (Seefeldt, 2021; doi.org/10.15784/601441).

## 1 Introduction

The in-situ measurement of precipitation in Antarctica is an exceedingly difficult task. It is challenging to measure precipitation in Antarctica due to the relatively small amount of precipitation (Palerme et al. 2014), the difficulty in distinguishing between falling snow (precipitation) and blowing snow (Souverinjns et al. 2018), and the need to do all of this in a remote environment requiring a low-power and autonomous measurement system. Although there are many difficulties
in the measurement of precipitation in Antarctica, it is still important that it is accurately measured given it is the dominant term in the surface mass balance (SMB) of the Antarctic Ice Sheet (AIS). The AIS contains the largest reservoir of freshwater on Earth and it is estimated that approximately 58 m of global mean sea level (GMSL) equivalent is stored in the AIS (Fretwell et al., 2013). The trend in the AIS mass balance is therefore a key in understanding the global impacts due to a warming climate. Medley and Thomas (2019) found that there has been a positive trend in snow accumulation on the AIS,
based on ice core records and reanalysis data, and that this increase in snow accumulation has reduced the twentieth century

GMSL rise due to the warming climate. This snow accumulation is referred to as the difference between total precipitation and losses due to evaporation/sublimation, wind-driven redistribution of snow, and meltwater runoff. Despite the significance of precipitation in the Antarctic climate system, and the broader Earth climate, there remains a lack of direct observations, understanding of the precipitation processes and links to the other key components of the AIS and SMB.


Currently, a large amount of what is known about precipitation and snow accumulation for Antarctica is from numerical model studies. The investigations range from using numerical weather prediction (NWP; e.g. Monaghan et al., 2005), reanalyses (e.g. Bromwich et al., 2011; Nicolas and Bromwich, 2011; Palerme et al., 2017; Wang et al., 2020b), earth system models (ESMs; e.g. Palerme et al., 2016; Lenaerts et al., 2016; Fyke et al., 2017; Wang et al., 2020a), and regional climate models (RCMs; e.g. Ligtenberg et al., 2013; Lenaerts et al., 2017). The combination of these models has provided a much greater understanding of the variability, trends, and mechanisms of precipitation over Antarctica. Medley et al. (2013) provides a verification of global and atmospheric models using air-borne radar and ice-core observations. However, there is an ongoing need for the direct measurement of precipitation and snow accumulation to provide an evaluation of the numerical models and insights into the uncertainties that are critical in understanding the AIS and GMSL in a warming climate.

Remote sensing studies of precipitation have been conducted using observations by CloudSat to determine and characterize precipitation rates across Antarctica. Palerme et al. (2014) created a multi-year, model-independent climatology of precipitation across Antarctica. The results indicated a mean annual precipitation over the Antarctic ice sheet, north of 82° S, of 171 mm year$^{-1}$ spanning the years 2006 to 2011. A follow up study, Palerme et al. (2019), with refinements to the methodology placed the annual precipitation rate over Antarctica, north of 82° S, at 159 mm year$^{-1}$. Additional studies of precipitation over Antarctica using CloudSat include Milani et al. (2018), extending the analysis to over the Southern Ocean, and Lemonnier et al. (2020), looking at rates of precipitation across several geographic regions and an investigation of the three-dimensional characteristics of precipitation over Antarctica.


Past studies of snow accumulation at field sites have increased our understanding of the connection between precipitation, weather systems, and the accumulation at a ground-based site. Braaten (1997) installed an instrument system that dispersed microspheres onto the snow surface at fixed intervals, with subsequent snow profile investigations, to track the accumulation rates over time with a comparison to the meteorology observations. Eisen et al. (2008) cover a range of ground-based measurements of snow accumulation, including stakes, ultrasonic sounders, snow pits, and firn and ice cores to provide an understanding of snow accumulation over East Antarctica. Knuth et al. (2010) used snow height measurements from sonic ranging sensors installed at automatic weather station (AWS) sites on the Ross Ice Shelf to track changes in snow height with the corresponding meteorological measurements to investigate precipitation and horizontal snow transport on surface accumulation. Cohen and Dean (2013) used the snow height measurements from the AWS on the Ross Ice Shelf for a study

comparing the changes in snow height to that of events in reanalyses. A lacking component in all of these surface-based measurement studies is the direct capturing of the precipitation in real-time with the precipitating event.

There is a significant need for direct measurements of precipitation to better evaluate the quality of the results in the numerical modeling studies and to be able to constrain the uncertainty in estimates of precipitation, snow accumulation, and the SMB of the AIS. A meteorological observatory was established at Princess Elisabeth base in East Antarctica (Gorodetskaya et al., 2015) for the observation of clouds and precipitation. The ground-based observations include a ceilometer, infrared pyranometer, and a vertically profiling precipitation radar (Micro Rain Radar) starting in 2010. The results from this project have provided insights into cloud and precipitation properties, with a focus on atmospheric rivers that produce significant precipitation events for East Antarctica (Gorodetskaya et al., 2014). A collection of precipitation sensors was installed at Rothera Station on the Antarctic Peninsula for an evaluation of the ability of the instruments to capture precipitation in Antarctica (Tang et al. 2018). Two tipping-bucket precipitation gauges, with heating elements, and three laser-based instruments, also known as disdrometers, were setup for 11 months in 2015-16. Only one of the tipping-bucket precipitation gauges had a wind shield and it was a single-Alter wind shield. The results were limited by being installed at a base, with no limits on power, and on the Antarctic peninsula, in contrast to over the expansive AIS. The recent APRES3 (Antarctic Precipitation, Remote Sensing from Surface and Space; Grazioli et al., 2017) field campaign setup an assortment of instruments at Dumont d'Urville station in East Antarctica for a focused effort on the monitoring of precipitation in terms of the collection and falling snow profile characteristics and microphysics. APRES3 had an intense measurement campaign for the austral summer 2015-16, with some of the instruments ongoing since then (Genthon et al., 2018). The instruments included a polarimetric radar (MXPol), a Micro Rain Radar (MRR), a weighing precipitation gauge (OTT Pluvio$^2$), and a Multi-Angle Snowflake Camera (MASC). The results of this field campaign have been used in several studies including evaluating CloudSat data (Lemonnier et al., 2019), reanalysis data (Grazioli et al., 2017), and a RCM (Vignon et al., 2019).

The Antarctic Precipitation System (APS) project, a collaboration between the University of Colorado Boulder (CU-Boulder) and the National Center for Atmospheric Research (NCAR), developed and installed precipitation measurement systems for Antarctica. The objective of the APS project was to install instruments for year-round, in situ measurement of precipitation at remote regions in Antarctica with low-power and autonomous operation. Four APS systems were installed for two years in the northwest region of the Ross Ice Shelf and collected year-round data. A description of the instrumentation and site infrastructure for each of the four sites is presented in section 2. Section 3 follows with a summary of the efforts during the field seasons with the installation, maintenance, and removal of the APS sites. The data collection, quality control, and data products produced by the APS project are covered in section 4. Section 5 provides a sample data analysis, including the data plots that are a part of the data repository, section 6 is a discussion the lessons learned, sections 7 and 8 cover data and code availability, and conclusions are covered in section 9.

## 2 Instrumentation and Installations

The instruments for the APS sites were designed based on the experience of solid precipitation measurement accrued at NCAR, associated with the World Meteorological Organization Solid Precipitation Intercomparison Experiment (WMO-SPICE; Nitu et al., 2018) project, and other precipitation-focused field experiments (Rasmussen et al., 2012). The knowledge and experience in solid precipitation measurement from NCAR was combined with lessons learned on the deployment of meteorological instruments in Antarctica from an association with the University of Wisconsin Antarctic AWS project

(Lazzara et al, 2012). The APSs were installed at four sites across the northwest Ross Ice Shelf (Fig. 1). All sites were confined to this region to minimize the necessary field logistics and support to complete the installations, maintenance, and removal of the APS sites. Table 1 provides the site location information for these four sites. Three APS sites were classified as standard installations with a consistent set of instrumentation across all sites. A fourth site was classified as the premier site as it had the standard collection of instruments and an expanded collection of instrumentation for additional experiments.

Table 2 provides a summary of the instrumentation at the standard sites and the additional instrumentation for the premier site. Figure 2 shows the APS installation at the standard APS site, Phoenix Airfield.

|  | Alexander Tall Tower | Lorne | Phoenix | Willie Field |
|---|---|---|---|---|
| Latitude | 79° 00.39' | 78° 11.35' | 77° 56.86' | 77° 52.05' |
| Longitude | 170° 44.1' | 170° 01.9' | 166° 45.5' | 166° 55.1' |
| Elevation | 55 m | 45 m | 10 m | 9 m |
| Install Date | 30 November 2017 | 29 November 2017 | 26 November 2017 | 27 November 2017 |
| Removal Date | 16 November 2019 | 15 November 2019 | 12 November 2019 | 20 November 2019 |

**Table 1: Latitude, longitude, elevation, installation date, and removal data for APS sites.**

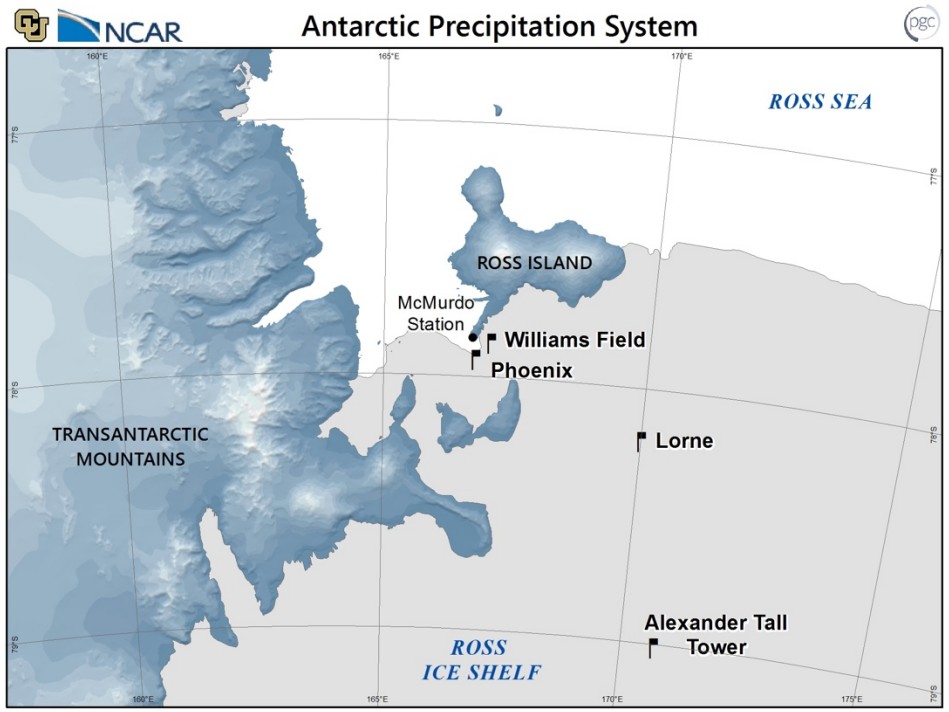

**Figure 1: Geographic map indicating the site locations for the installed APSs in the Ross Island region of the northwest Ross Ice Shelf in Antarctica. Figure was provided by the Polar Geospatial Center.**

| | Alexander Tall Tower | Lorne | Phoenix | Willie Field |
|---|---|---|---|---|
| **Precipitation** | OTT Pluvio$^2_a$ | OTT Pluvio$^2_a$ | OTT Pluvio$^2_a$ | OTT Pluvio$^2_a$ |
| | | | | OTT Pluvio$^2_{bc}$ |
| | | | | OTT Pluvio$^2_{de}$ |
| **Wind Speed** | Vaisala WAA151 | Vaisala WAA151 | Vaisala WAA151 | Vaisala WAA151 |
| | | | | Vaisala WAA151$_f$ |
| **Air Temperature** | Campbell Sci. SR50AT | Campbell Sci. SR50AT | Campbell Sci. SR50AT | Campbell Sci. SR50AT |
| **Snow Height** | Campbell Sci. SR50AT | Campbell Sci. SR50AT | Campbell Sci. SR50AT | Campbell Sci. SR50AT |
| **GPS-IR** | Trimble NetR9 | Alert Geomatics Resolute Polar | Septentrio PolaRx5 | Septentrio PolaRx5 |
| **Particle Count** | ETI Optical Precipitation Detector | ETI Optical Precipitation Detector | ETI Optical Precipitation Detector | ETI Optical Precipitation Detector |
| **Disdrometer** | Thies Laser Precipitation Monitor$_g$ | OTT Parsivel$^2_{gh}$ | OTT Parsivlel$^2_g$ | Thies Laser Precipitation Monitor$_g$ |
| **Pyranometer - Down** | – | Kipp & Zonen CMP3 | Kipp & Zonen CMP3 | – |
| **Pyranometer - Up** | Kipp & Zonen CNR4 | – | Kipp & Zonen CMP3 | Kipp & Zonen CNR1 |
| **Pyrgeometer - Down** | Kipp & Zonen CNR4 | Kipp & Zonen CGR3 | Kipp & Zonen CGR3 | Kipp & Zonen CNR1 |
| **Pyrgeometer - Up** | Kipp & Zonen CNR4 | Kipp & Zonen CGR3 | – | Kipp & Zonen CNR1 |
| **Field Camera** | Campbell Sci. CCFC | Campbell Sci. CCFC | Campbell Sci. CCFC | Campbell Sci. CCFC |

a = Installed inside a double-Alter wind shield at an orifice height of approximately 3.4 m

b = Installed insde a double-Alter wind shield at an orifice height of approximately 1.9 m during the first year

c = Installed insde a modified single-Alter wind shield at an orifice height of approximately 2.4 m during the second year
d = Installed insde a DFIR wind shield, with an inner single-Alter wind shield, at an orifice height of approximately 1.9 m during the first year

e = Installed without wind shielding at an orifice height of approximately 2.5 m during the second year

f = Installed at a height of approximately 1.8 m

g = Installed during the first year but no data collected due to hardware/software issues.

h = Removed for the second year due to instrument failure

**Table 2: Instruments installed at each APS site including instrument manufacturer and model. The notes provide specifications of the instrument that is different from the description in the text and unique to the indicated site.**

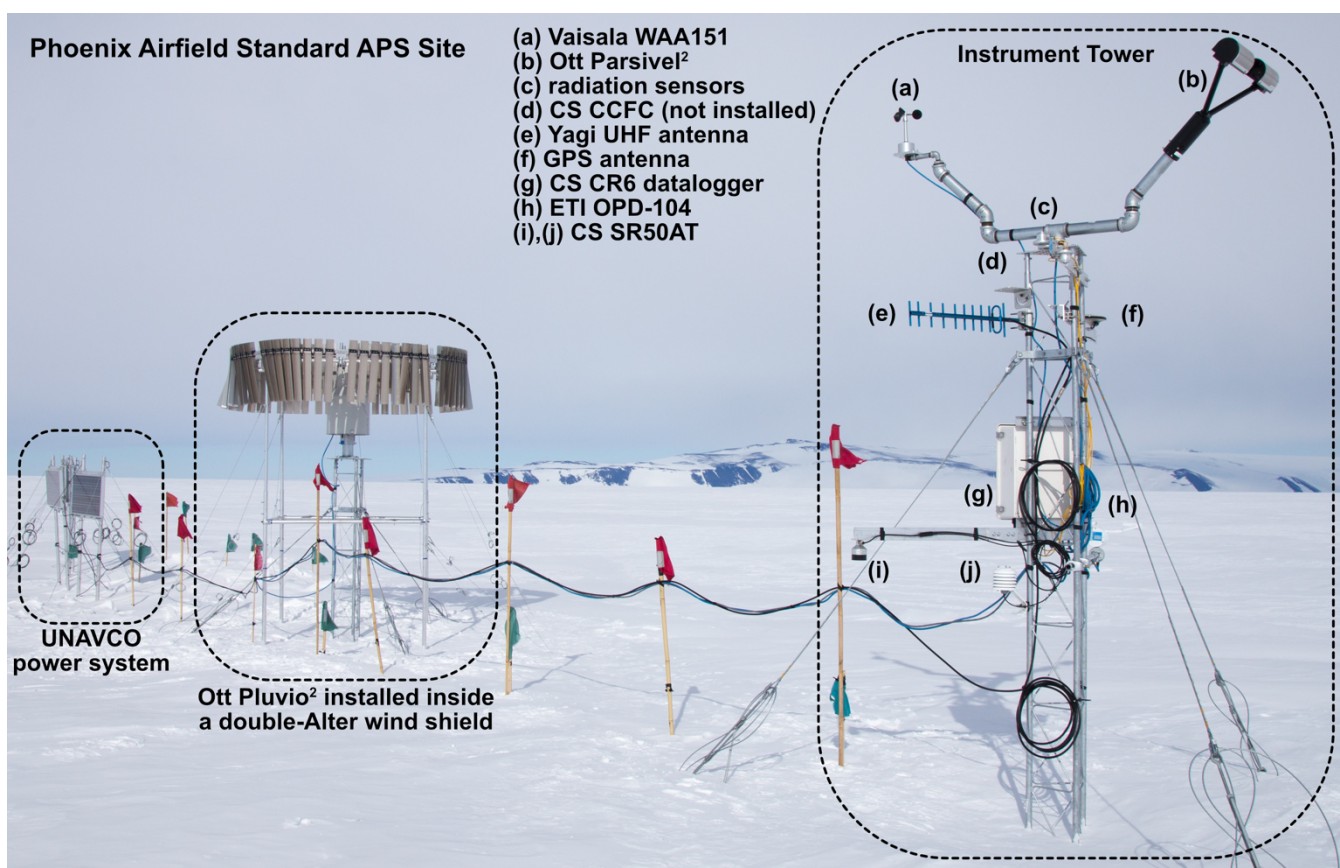

**Figure 2: A complete APS standard site installation at the Phoenix Airfield site at the completion of the installation in November 2017. The UNAVCO power system is at the far left, the weighing precipitation gauge installed inside the wind shield at the center, and the instrument tower is on the right.**

## 2.1 Site Infrastructure

The infrastructure for the APS sites was mostly uniform across all four sites. The exception to this is the Willie Field premier site that included the additional instrumentation for cross-instrumentation evaluation, which will be detailed in a later section. An APS site was composed of three components: a precipitation gauge with wind shield, an instrument tower, and a UNAVCO power system. Rohn 25G tower sections were used for the precipitation gauge and the instrument tower. The top of the instrument tower was approximately 3.3 m above the surface at the time of installation. The instrument tower and the precipitation gauge were separated by a distance of approximately 10 m. The bottom of both towers was installed approximately 0.6 m beneath the surface and packed in snow to provide a firm base for the installation. Each of the Rohn 25G towers were then anchored by three 0.0063 m (¼ inch) wire rope guys, which were anchored into the snow using snow boards, to provide stability in strong winds.

The operation of the instruments, data collection, processing of observations, data storage, and communications was handled by a Campbell Scientific CR6 datalogger at all four sites. The CR6 was enclosed in a fiberglass weatherproof enclosure and mounted on the instrument tower. A MicroSD card was included with the CR6 datalogger for storage of the data between field visits. Communications with the datalogger were accomplished with an internet connection between Boulder, Colorado and McMurdo Station. UHF radio communications, using an Intuicom EB-1 ethernet bridge radio with line-of-sight yagi antennas, allowed communications between each APS site and McMurdo Station. The communications link allowed for year-round, near real-time access to the datalogger for data retrieval and uploading updated datalogger algorithms.

The power to the datalogger and instruments was provided by a UNAVCO power system. The power system was designed to provide 3W of continuous power during the long polar night. The power system is comprised of two 90 W solar panels, 16 12V 100 Ah batteries, and a charge controller to handle the charging of the batteries, during summer months, and distribution of power. The batteries and charge controller were placed in two weather resistant enclosures and buried beneath the snow surface to minimize the temperature variations on the batteries during the course of the year. The power systems operated successfully throughout the duration of the project, except for insufficient capacity of the batteries to power the Lorne and Alexander Tall Tower APS sites through the polar night.

## 2.2 Standard Sites

The OTT Pluvio$^2$ precipitation gauge was used for the measurement of precipitation. The Pluvio$^2$ uses a high-precision load cell, in combination with algorithms designed to compensate for the wind and temperature, to produce measurements of amount and intensity of precipitation using a weight-based technology. The collection area of the Pluvio$^2$ was 400 cm$^2$ at the Alexander Tall Tower, Willie Field, and Lorne APS sites and 200 cm$^2$ at the Phoenix Airfield site. The Pluvio$^2$ was mounted on a Rohn 25G top plate at the top of a Rohn 25G tower, at a height of approximately 2.4 m above the surface. This resulted in the orifice opening of the Pluvio$^2$ at a height of approximately 3.4 m at time of installation. (The actual height of the instruments varied over the duration of the project due to the increasing height of the snow surface, which is common to this region.) The elevated height of the Pluvio$^2$ was chosen to minimize the interaction of the collection of precipitation with the blowing snow near the surface. A custom fabricated double-Alter wind shield was installed surrounding the Pluvio$^2$ to slow the horizontal transport of snow by the winds and to increase the collection rate of the precipitation gauge. The customizations to the double-Alter wind shield were made to increase the functionality and durability of the wind shield in the harsh Antarctic climate conditions. The top of the wind shield was installed at the same height as the opening of the Pluvio$^2$. A customized base of aluminum piping was fabricated to install and support the wind shield at the elevated height of the precipitation gauge. The Pluvio$^2$ and wind shield is at the center of the Fig. 2.

The snow surface height was measured with a Campbell Scientific SR50AT sonic ranging sensor with temperature probe. The SR50AT operates by emitting an ultrasonic pulse and measuring the elapsed time between the emission and the return of

the pulse. The snow height is calculated based on this elapsed time and a temperature corrected speed of sound. The
SR50AT sonic ranging and temperature sensors were installed on the instrument tower at a height of approximately 1.0 m, at
the time of installation, with slight variations by site. In addition to being used for correcting the speed of sound, the
temperature sensor was also used to provide continuous measurements of air temperature. Snow height was also measured
using a GPS receiver and the method of GPS-IR, detailed later in this paper.

Additional instrumentation was installed on the instrument tower to characterize the atmospheric conditions to better
understand and interpret the measurements by the Pluvio[2] and SR50AT. A Vaisala WAA151 3-cup anemometer was
installed at approximately the same height as the opening of the Pluvio[2] to provide a measurement of wind speed at gauge
height. An ETI Optical Precipitation Detector (OPD-104) was installed at a height of approximately 1 m above the surface.
The OPD-104 provided a continuous, robust, and low-power measurement of a count of particles in the air moving past the
sensor (either by falling or blowing through the LED light beam). This measurement was used to characterize occurrences of
precipitation and/or blowing snow. Whenever the OPD-104 was measuring more than 30 counts in 6 seconds, for a sustained
two minutes, the conditions would be classified as an "event". The classification of an "event" triggers additional
instruments to be used, as detailed below. A Campbell Scientific CCFC Field Camera was installed near the top of the
instrument tower to provide a visual record of the conditions of the precipitation gauge and wind shield during events, as
classified by the OPD-104. The CCFC Field Camera included infrared LEDs providing operation throughout the long polar
night. A laser disdrometer was installed at approximately the height of the opening of the precipitation gauge on the
instrument tower to provide a measurement of particle size of the blowing snow and precipitating snow. Two different
models of disdrometers were used with an OTT Parsivel[2] installed at Phoenix Airfield and Lorne APS sites and a Thies
Laser Precipitation Monitor (LPM) installed at Willie Field and Alexander Tall Tower APS sites. The disdrometers were
installed at a 45° angle to the horizontal (Fig. 2) to improve the ability to capture the horizontally moving blowing snow. The
instrument algorithms for fall velocities, precipitation rates, and kinetic energy are not able to accommodate this tilt of the
instrument. This was not considered to be an issue with the APS installation as the primary use of the disdrometers was to
capture particle counts and sizes to better attempt to distinguish between blowing and precipitating snow. The disdrometers
were in operation only during events to minimize the higher power consumption of these instruments. Lastly, radiation
sensors were installed at the APS sites for measuring downwelling and upwelling shortwave and longwave radiation. A Kipp
& Zonen CNR1 net radiometer was installed at Willie Field, and a CNR4 net radiometer was installed at Alexander Tall
Tower, providing measurements of downwelling shortwave and longwave and upwelling longwave radiation. Two Kipp &
Zonen CMP3 pyranometers and a Kipp & Zonen CGR3 pyrgeometer were installed at Phoenix Airfield for measurements of
shortwave downwelling and upwelling radiation, and longwave downwelling radiation. Lorne had a slightly different set of
instruments with one CMP3 pyranometer and two CGR3 pyrgeometers for measurements of downwelling shortwave, and
downwelling and upwelling longwave radiation.

## 2.3 Premier Site

The Willie Field site was classified as the premier site for the APS project. Figure 3 is a photo of the Willie Field APS premier site at the time of installation. The premier site had all of the instrumentation installed as the standard sites and it included two additional Pluvio[2] precipitation gauges. The experiment to be conducted with these gauges changed after the first year for reasons that will be explained below in the section covering the Season 2 field work. In this section, the original configuration and rationale for the premier site at the outset of the project will be described. One of the additional Pluvio[2] precipitation gauges was installed at a height of approximately 1.9 m, inside a Double Fenced Automated Reference (DFAR) wind shield. A DFAR is the use of a Double Fence Intercomparison Reference (DFIR) wooden fence, as the wind shield, in combination with an automatic precipitation gauge inside a single-Alter wind shield, as defined by the WMO SPICE report (see Fig. 3.4 in Nitu et al., 2018). The DFIR-fence is an octagonal, vertical, double-fence inscribed in circles of 12 m and 4 m in diameter. The DFAR wind shield is considered the WMO automated standard for the measurement of solid precipitation (Nitu et al., 2018). The installation of a DFAR at all sites would have been costly and prohibitive to install with limited field installation time. Therefore, a DFAR was installed at Willie Field to provide a comparison to the measurement of precipitation in the double-Alter wind shield. The goal was to develop a transfer function between the measurement of precipitation with the double-Alter wind shield and the DFAR wind shield that could then be applied to the other APS sites to correct the gauge measurements for wind under catch. An additional Pluvio[2] precipitation gauge at Willie Field was installed at a height of 1.9 m, matching that of the DFAR, inside a double-Alter wind shield. This side-by-side comparison of the DFAR to a double-Alter shielded precipitation gauge was designed to minimize the impacts that different heights of the gauge installations could have on precipitation collection. An additional Vaisala WAA 151 3-cup anemometer was also installed on the instrument tower at the same height (1.9 m) of the opening of the two additional Pluvio[2] precipitation gauges to determine the wind speed at their gauge heights. A UNAVCO 5W power system was installed at the Willie Field site to accommodate the additional instrumentation at this site. The 5W power system is the same as the 3W except with six additional 12-V 100 Ah batteries in a third weather-tight enclosure buried beneath the surface, similar to the two other enclosures.

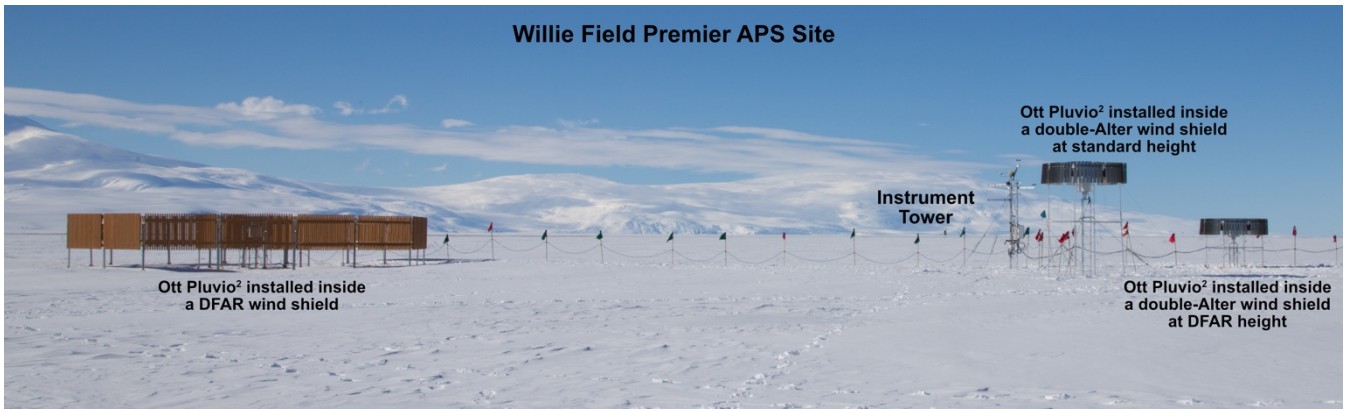

**Figure 3: The APS premier site installation at the Willie Field site at the completion of the installation in November 2017. The precipitation gauge installed inside a double-fenced automated reference (DFAR) shield is on the left and the precipitation gauge installed inside a double-Alter wind shield, at the same height as the DFAR precipitation**
**gauge, is on the right. The instrument tower and precipitation gauge installed in the double-Alter wind shield are similar to that of the standard sites.**

## 2.4 GPS-IR

A novel method of measuring snow height through the interferometric pattern of Global Positioning Satellites (GPS) signals was included in the project to provide for an additional measurement of snow height (Larson et al., 2009; Larson et al., 2015;
Siegfried et al., 2017; Larson et al., 2020). A GPS receiver was included in the instruments at each APS site to complete the GPS interferometric reflectometry (GPS-IR) measurements. This additional measurement of snow height at the APS sites provided an opportunity to demonstrate the capabilities of GPS-IR for studies involving precipitation, snow accumulation, and SMB of the AIS (see references for GPS-IR). GPS-IR operates on the principle that GPS signals at low elevation angles are reflected off of the snow surface. This reflected signal creates an interference pattern with the direct GPS signal to the
geodetic antenna (see Fig. 1 in Larson et al., 2020). This interference pattern has a characteristic signal-to-noise ratio that can be used to retrieve the height of the GPS antenna above the snow surface. This pattern of snow height measurements by GPS-IR is similar to that of the observations from the sonic ranging sensor over the range of weeks to months. However, there are different characteristics in how the values are measured and in the pattern of observations over hours to days. For example, the sonic ranging sensor has a footprint with a radius of tens of centimeters and the GPS-IR has a footprint with a
radius of tens of meters. The sonic ranging sensor takes measurement every minute while the GPS-IR processing creates a daily average. The result is a much more uniform measurement representative of the larger APS site rather than the snow surface directly beneath the sonic ranging sensor that can be influenced by migrating drifts and sastrugi. A study currently in preparation compares the measurements from the GPS-IR, sonic ranging sensor, precipitation measured by the precipitation gauge, and wind speed for an improved characterization of snow accumulation at a given site. The results will also be
beneficial to better understand previous studies that relied on similar sonic ranging sensors for snow height measurements (e.g. Eisen et al., 2008; Knuth et al., 2010; Cohen and Dean, 2013).

## 3 Field Work

### 3.1 Season 1: November 2017

The goal for the first season was the installation of the four APS sites to provide for the collection of two years of
observations over the duration of the project. The first field season lasted from 31 October to 1 December 2017. Much of the time in the field was spent at the two sites nearby McMurdo Station, Willie Field and Phoenix Airfield, as they allowed for ease of access using a wheeled, or tracked, light vehicle truck. This provided the ability to refine and adapt the installation

hardware and methods for the unique Antarctic environment. These modifications and improvements in the field installation resulted in the entire Lorne APS site being installed in under seven hours with four field personnel on the second to last day

of the field season. The installation of the DFAR at the APS premier site, Willie Field, was also a success and required a full day for installation with five field personnel. All sites were operating at the conclusion of the field season. An additional 4-6 weeks of remote work, depending on the site, after the completion of the field season were required to address issues with the measurement and communication algorithms before the commencement of recording observations from the four APS sites.

**3.2 Season 2: November 2018**

The activities for the second field season concentrated on the maintenance and repairs for the four APS sites. The first year was an overall success in terms of retrieving measurements of precipitation, especially when factoring in the frequent difficulties encountered with typical instrument deployments in Antarctica. There were minor issues with the instruments and algorithms that needed further refinements to improve the results and quality of observations for the second year. The

second field season lasted from 5 November to 7 December 2018. One of the major repairs was made to the aluminum piping base for the wind shield at the Phoenix Airfield APS site, which became partially unassembled during a windstorm in early September 2018. All joints in the aluminum piping at this site, as well as the other three sites, were reinforced with bolts during the second field season. Measurements from the disdrometers at all four sites were unsuccessful during the first year due to a combination of hardware and software issues. These were sorted out during the second field season with the

disdrometers working at all sites, except for Lorne site where the Parsivel[2] had to be removed due to a hardware failure. Maintenance at each of the sites included repositioning some of the instruments on the instrument tower to new heights and, when necessary, fixing or tightening any instrumentation hardware mounts.

The most significant negative outcome from the first year was that the DFAR at the premier site became fully buried due to

drifting and blowing snow. Observations from the Pluvio[2] precipitation gauge indicate that in mid-March 2018 the snow surface reached the height where the drifting snow was able to enter into the opening and fill the precipitation gauge. It is uncertain as to when the DFAR became fully buried during the austral winter. The dome of snow that was created, as a result of the burying of the DFAR, resulted in the snow height increasing dramatically (approximately 1-2 m) at the Willie Field APS site, including the instrument tower and the two Pluvio[2] precipitation gauges installed in the double-Alter wind shields.

Past measurements at the nearby AWS site indicate that this site typically has an annual increase in snow height of approximately 0.4 m. The entire DFAR installation was removed from the site with the associated Pluvio[2] precipitation gauge being re-installed in an unshielded configuration. The Pluvio[2] precipitation gauge with a double-Alter wind shield at the lower height was raised and modified by removing the outer shield and creating a single-Alter wind shield configuration. The result of the field work was a modified experimental setup at the premier site for the second year with three Pluvio[2]

precipitation gauges having different configurations of wind shielding: double-Alter, single-Alter, and unshielded. The lower

instruments (e.g. OPD-104, SR50AT, lower WAA151) were raised on the instrument tower due to the increase in snow height at this site. Figure 4 is a photo of the reconfigured premier Willie Field APS site for the second year of data collection.

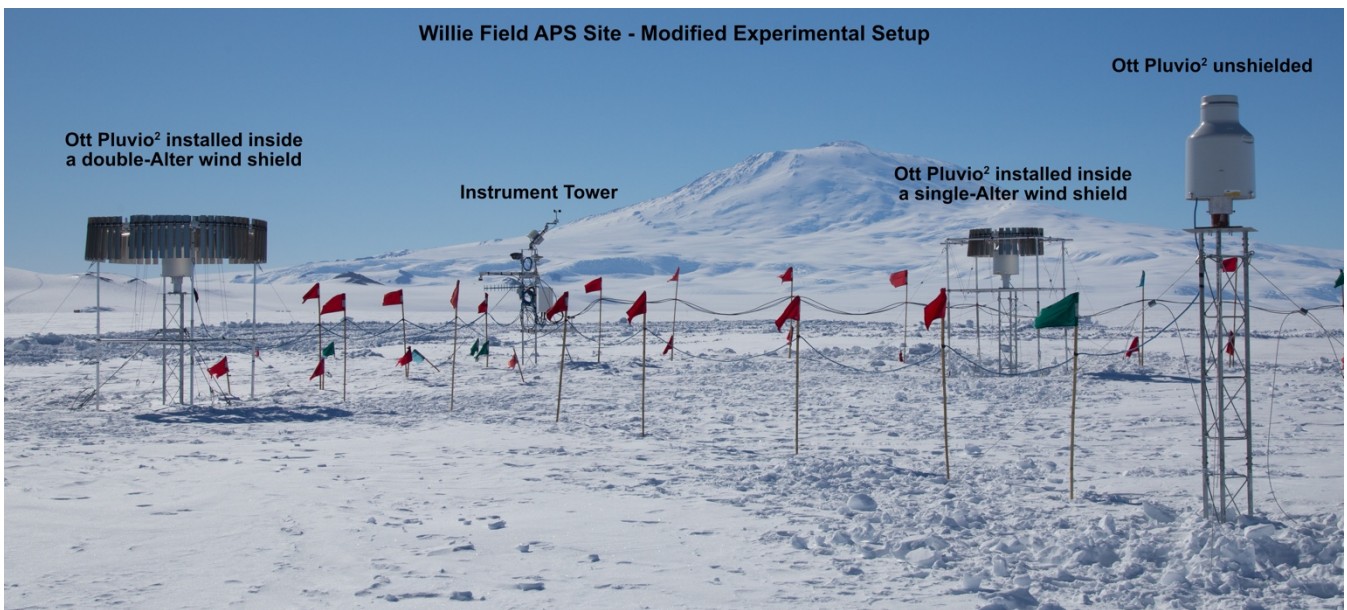


**Figure 4: The modified experimental setup at the Willie Field site for the second year, at the completion of the modifications in November 2018. The three precipitation gauges are installed in a double-Alter wind shield, single-Alter wind shield, and unshielded from left to right in the photo.**

### 3.3 Season 3: November 2019

The third field season involved the removal of all instruments and hardware marking the completion of the field deployment for the project. The field season lasted from 7 November to 3 December 2019. The instruments, hardware, and power systems for each site were successfully removed without any issues. Each APS site was surveyed and assessed for the conditions of the instruments and hardware for lessons learned to be applied to potential future deployments of the APS project. One ongoing issue to be addressed is the occurrence of capping within the opening of the Pluvio$^2$ precipitation

gauge. Capping is the occurrence of snow accumulation on the rim of the gauge leading to partial or complete blockage of the orifice (Nigu et al., 2018). At differing points during the austral winter, each of the Pluvio$^2$ precipitation gauges stopped measuring precipitation due to the orifice of the gauge becoming blocked with snow. This issue is indicated in the observations with the measurement of liquid water equivalent (LWE) being constant for days and extending into weeks. On several occasions over the two years, the observations at a site would indicate a sudden increase in the precipitation

measurement due to the accumulation of snow/ice on the inside of the orifice being loosened and falling onto the Pluvio$^2$ load cell in a large dump.

## 4 Quality Control and Data Processing Section

The data from the four APS sites were collected remotely in Boulder, Colorado. In general, the data was retrieved hourly to a local data server using the Campbell Scientific LoggerNet software. The data was also stored on a MicroSD card installed in
the Campbell Scientific CR6 datalogger, which provided a second source of data to fill any gaps that occurred due to issues with the radio communications. This was a rare event. The observations were collected in three intervals: 6 seconds, 1 minute, and 5 minutes. The 1-minute data is considered the primary data source and was the data which underwent subsequent quality control (QC) and sharing for community use. Initial processing of the 1-minute data was completed using a csh script to create daily text files (obs1min) of the collected data from the larger LoggerNet text files. This process was
repeated for the duration of the two years that the APS sites were in operation. The 6-second data was collected to capture a higher temporal resolution providing more information on the performance of the Pluvio$^2$ precipitation gauges. The 6-second observations did not undergo QC processing, having only been used and archived for instrument characterization, and are not included in the data repository. The 5-minute data are not continuous and were only recorded for occurrences when the APS site was considered to be experiencing an "event", as determined by the OPD-104 measurements. The snow height and
air temperature measurements from the SR50AT were included in the 5-minute observations for the first three months of the first year at Phoenix Airfield and Willie Field, and the entire first year at Lorne and Alexander Tall Tower. There were no measurements from the disdrometers during the first year, as was discussed earlier. Single text files for the first year and for each site were created containing the 5-minute observations (obs5min) and are posted without QC processing. The disdrometer data in the second year was recorded with 1-minute resolution, active only during events, and stored in a single
file for each site. Table 3 provides a summary of the observations that were collected by time resolution, site, and for which years the observations were completed.

The 1-minute data underwent QC processing at CU-Boulder for use in the scientific analyses and for posting of the data to the repository for community use. The QC processing, using an automated algorithm, was applied to the 1-minute data for
the Bucket NRT measurements from the Pluvio$^2$, the air temperature and snow height measurements from the SR50A, the particle count from the OPD, and wind speed from the WAA151 (see Table 3). The Bucket NRT values are the non-real-time measured values of bucket content, corresponding to the filtered weight value from the Pluvio$^2$. These five measurements are considered the primary observations for analysis of the results from the APS project. The automated QC processing was developed through iterations of manually viewing the observation dataset spanning the two years for each
APS site. Suspect and/or erroneous values were identified, and algorithms were developed to automatically remove the points in the data processing. The QC algorithms were unique to each site and instrument and included methods to remove

outliers beyond an established range, removing values beyond a threshold from a rolling mean, and removing measurements due to a faulty instrument (e.g. the OPD 104 being buried at Willie Field APS site). For example, at all sites the datapoint is removed when the absolute difference of the Bucket NRT value and a rolling mean of 10 minutes of observations is larger

than 10 mm LWE. Another example is the removal of wind speed observations of 0 m s$^{-1}$ for more than 180 minutes. This is an indication of the 3-cup anemometer being stuck, likely due to being frozen from riming or some other icing of the instrument. The result is an automated and objective QC processing developed using manual analyses and subjective criteria to create a clean dataset.

Images and movies were also captured of the APS sites using the CCFC Field Camera installed with the instruments. The camera captured 5-second videos during the first year at one-hour intervals during events. Photos and 5-second videos were captured during the second year at one-hour intervals during events, and a photo was captured every day at 00 UTC to indicate conditions at each site. The videos in the first year, and still images in the second year, were retrieved to the data server in Boulder, Colorado using the remote communications and LoggerNet software. However, it is the videos and images

stored on the MicroSD card that were used for the APS data archive. The videos were converted in post-processing from the CCFC Field Camera specified avi video format to a more user friendly mp4 video format and encoding. The images and videos were manually reviewed and all images and videos that had obstructed views, providing no usable information, were removed. The views might have been obstructed due to accretion of ice on the camera lens, extensive blowing snow in the field of view making the image useless, or a configuration error at Phoenix Airfield site that did not enable the infrared LEDs

during the first year. All of the useful images and videos are included in the repository.

The observations from the GPS receivers were collected by UNAVCO through either a UHF and internet link, similar to the APS communications, or an Iridium satellite link. The data processing and quality control for the GPS-IR observations are being handled by the Department of Geophysics at the Colorado School of Mines. They are completing the processing of

GPS-IR observations for the entire Antarctic continent, beyond the four APS sites. The data processing, documentation, and public release of the data is under development at this time.

Corresponding meteorology observations for each of the sites can be retrieved from the co-located University of Wisconsin AWS sites (Lazzara et al. 2012). The standard measurements available at each of the sites include atmospheric pressure,

wind speed, wind direction and relative humidity. Additional measurements are available at each site, highlighted by Alexander Tall Tower, which is an instrumented meteorological tower with a height of 30 m with six levels of measurements of wind, temperature, and relative humidity.

| Time Resolution | Instrument | Measurement | ATT | | LRN | | PHX | | WFD | |
|---|---|---|---|---|---|---|---|---|---|---|
| | | | 1 | 2 | 1 | 2 | 1 | 2 | 1 | 2 |
| **1 minute** | Pluvio | Accu RT-NRT | X | X | X | X | X | X | X | X |
| | | Accu NRT | X | X | X | X | X | X | X | X |
| | | Accu total NRT | X | X | X | X | X | X | X | X |
| | | Bucket RT | X | X | X | X | X | X | X | X |
| | | Bucket NRT | Q | Q | Q | Q | Q | Q | Q | Q |
| | SR50A | Snow Height | | Q | | Q | Q | Q | Q | Q |
| | | Temperature-Air | | Q | | Q | Q | Q | Q | Q |
| | WAA151 | Wind Speed | Q | Q | Q | Q | Q | Q | Q | Q |
| | OPD | Count | Q | Q | Q | Q | Q | Q | Q | Q |
| | | Path Status | X | X | X | X | X | X | X | X |
| | Radiation | Shortwave-Downward | - | - | X | X | X | X | - | - |
| | | Shortwave-Upward | X | X | - | - | X | X | X | X |
| | | Lonwave-Downward | X | X | X | X | X | X | X | X |
| | | Longwave-Upward | X | X | X | X | - | - | X | X |
| **Event Based (1 minute and 5 minute)** | SR50A | Snow Height | X | | X | | | | | |
| | | Temperature-Air | X | | X | | | | | |
| | Parsivel | Num. Part. Detected and Valid | - | - | | - | | X | - | - |
| | | Num. Part. Detected - All | - | - | | - | | X | - | - |
| | | Avg. Vol.Eq.Dia. (ved) of n Class | - | - | | - | | X | - | - |
| | | Avg. Part. Speed (ps) of n Class | - | - | | - | | X | - | - |
| | | Raw Data (Bins of Spd, Dia) | - | - | | - | | X | - | - |
| | LPM | Precipitation Amount | | X | - | - | - | - | | X |
| | | 1min Visibility in Precipitation | | X | - | - | - | - | | X |
| | | Number Particles | | X | - | - | - | - | | X |
| | | Number Particles LT 0.15 m/s | | X | - | - | - | - | | X |
| | | Number Particles GT 20 m/s | | X | - | - | - | - | | X |
| | | Number Particls LT 0.15 mm | | X | - | - | - | - | | X |
| | | Number Particles - NoHydro | | X | - | - | - | - | | X |
| | | Number Particles - Unknown | | X | - | - | - | - | | X |
| | | Number Particles - Classes 1-9 | | X | - | - | - | - | | X |
| | | Total Volume - Classes 1-9 | | X | - | - | - | - | | X |
| | | Raw Data (Bins of Spd, Dia) | | X | - | - | - | - | | X |
| **1 hr (EB)** | CCFC | Image | | Q | | Q | | Q | | Q |
| | | Video (5s) | Q | Q | Q | Q | Q | Q | Q | Q |
| **Daily** | CCFC | Image | | Q | | Q | | Q | | Q |
| | | Video (5s) | | Q | | Q | | Q | | Q |

Table 3: Data availability listed by measurement, site, and year. Measurements indicated by a Q indicate quality-controlled measurements, a X indicates non-quality-controlled measurements, and a dash indicates that the instrument was not installed at a site.

## 5 Plots and Sample Data Analysis

Time series of plots of the quality-controlled, 1-minute observations, of the five primary APS measurements, have been created with durations of 10-days, 1-month, and the entire observation period for each APS site. The plots for the Willie Field premier site include the Bucket NRT measurements from the additional Pluvio² instruments, and the lower wind speed. All of these plots are included in the data repository to provide an overview of the APS data for interested users, as well as actual analyses of individual events.


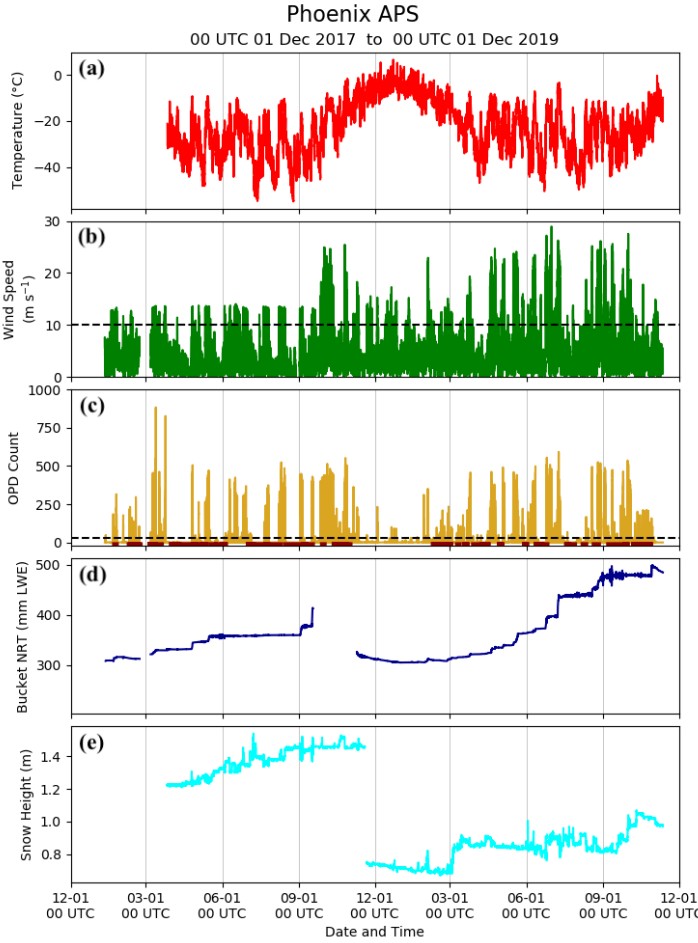

**Figure 5: Time series plots of the quality-controlled data from Phoenix Airfield APS site spanning from December 2017 to November 2019. Plotted data include temperature (a), wind speed (b), particle count (c), Bucket NRT (d; see text for a description), and snow height (e). The dashed line in (b) indicates a wind speed of 10 m s¹, which is the**

**approximate speed when there is blowing snow at the surface. The dashed line in (c) indicates a particle count of 30, which when sustained is classified as an "event" and triggers additional observations.**

Figure 5 is a plot of the quality-controlled, 1-minute observations from Phoenix Airfield APS site spanning the entire period of data collection. The plot covering the entire data record for each site provides an overview of the performance of the instruments and broad characterizations of the data. Figure 5a indicates the pattern of temperature at the site over the course of installation, once the SR50A data were added to the 1-minute data files starting in March 2018. The Bucket NRT data (Fig. 5d), shows the progression of precipitation accumulation during the first year until late May when the Pluvio$^2$ becomes capped and additional precipitation is not collected. In early September 2018, with the gauge warming due to solar radiation, the capping collapses indicated with the spike in the Bucket NRT values. That is followed with a data gap, which was the result of the disassembly of the wind shield base due to high winds during a storm. The Bucket NRT value decreases as the start of the second year after the removal of the snow collected in the bucket from the first year, during the field maintenance at the Phoenix Airfield site. There is then a steady event-by-event increase in the precipitation collection at Phoenix Airfield site until late June 2019 when the Pluvio$^2$ capping eliminates the collection of precipitation in the Pluvio$^2$. The Bucket NRT values are limited to event-by-event observations of precipitation for this dataset and APS deployment. This limitation is due to the capping of the Pluvio$^2$ instrument, not factoring in evaporation (loss) occurring in the bucket, and the emptying of the bucket at the sites in November 2018. The SR50AT measures a distance from the sensor to the surface, which is indirectly a measurement of snow height (Fig. 5e). In order to better correspond to increasing snow height, the value from the SR50AT is subtracted from a constant, such as 2 m for Phoenix Airfield APS, in order to indicate a positive trend for increasing snow height. The break in the snow height data in November 2018 is the result of the repositioning of the SR50AT sensor on the instrument tower during the 2018 field season.

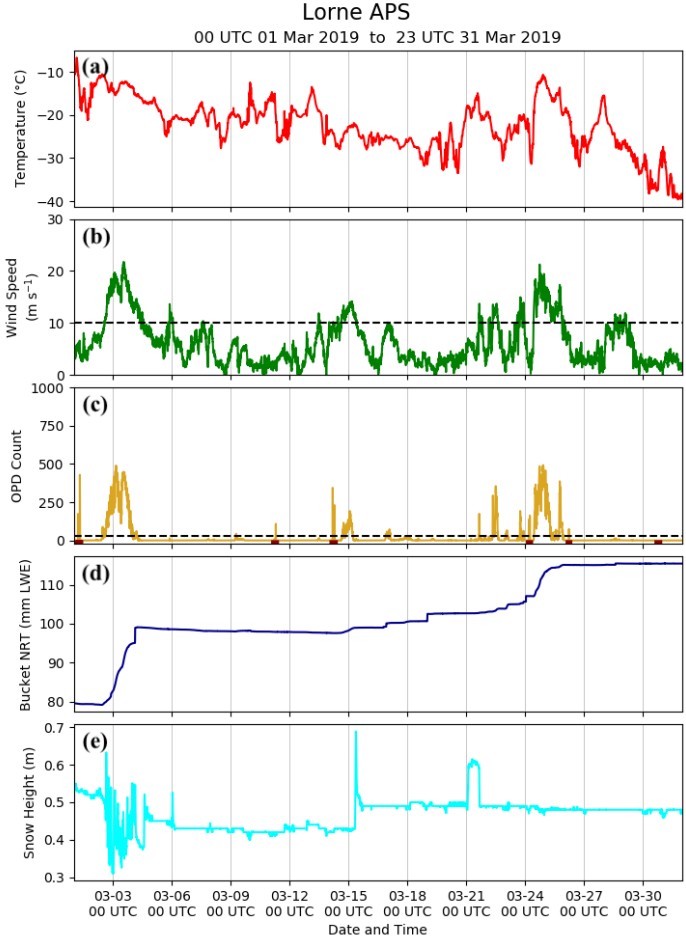

**Figure 6: The same as Fig. 5 but for Lorne APS site and March 2019.**

Figure 6 shows the analysis of the quality-controlled observations for March 2019 at Lorne APS site. The 10 m s$^{-1}$ line is highlighted in Fig. 6b to indicate the approximate wind speed when there is blowing snow near the surface. This threshold of blowing snow is verified with the particle counts from the OPD (Fig. 6c) increasing due to the blowing snow. The 30-count line is highlighted in Fig. 6c as this is the level of particles measured by the OPD classifying as "event" and the additional measurements by the CCFC Field Camera and disdrometer. The Bucket NRT measurements indicate the collection of precipitation during a big event at the start of the month, some smaller accumulations in the middle of the month, and another large event towards the end of the month. There are also noticeable large and sharp increases in the Bucket NRT, such as on 4 March 2019, when it appears that snow and ice that had accumulated on the surface inside the orifice, falls into the bucket in a dump. Figure 6e shows the large variability in snow height during high wind speed events and the lack of correlation of the precipitation to the accumulated snow height. This feature will be highlighted in a forthcoming study currently in preparation.

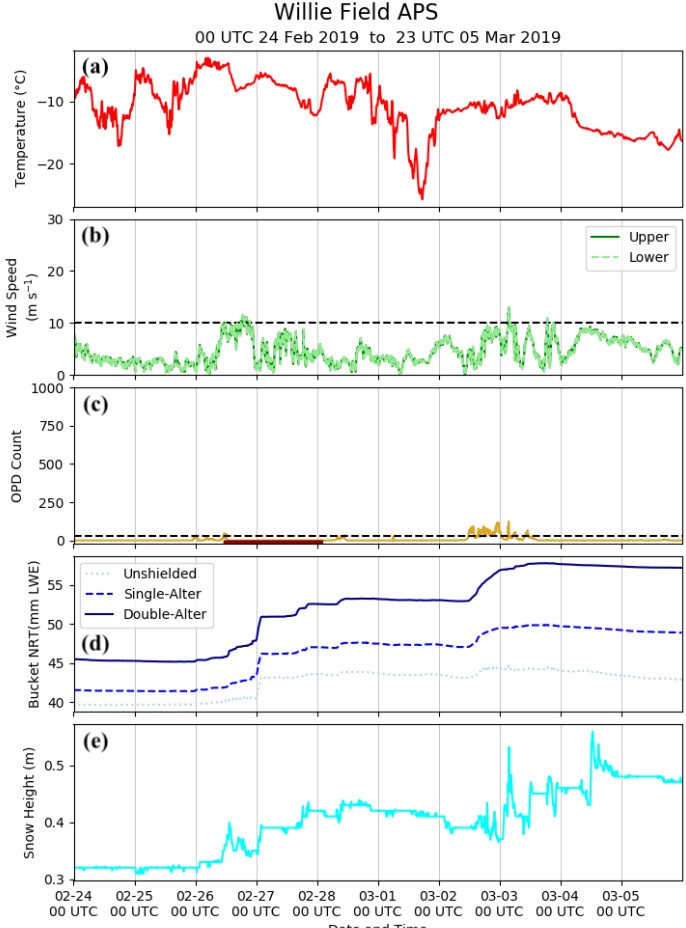

Figure 7: The same as Fig. 6 but for Willie Field APS site for 24 February through 5 March 2019.

Specific precipitation events are better analyzed using 10-day plots from each APS site. Figure 7 shows a 10-day plot of APS observations from 24 February 2019 through 5 March 2019. Similar features as described with Fig. 5 and Fig. 6 can be applied to the 10-day plots. The dark red line that is a part of Fig. 7c indicates the times when the OPD instrument is indicating that the path is blocked for the detection of particle counts. Figure 7d shows the results of the modified experimental configuration at Willie Field APS site for the second year in comparing precipitation measurements Pluvio$^2$ instruments without a wind shield, and with single-Alter and double-Alter wind shield configurations. There are light winds throughout this 10-day period which results in a better correlation between the snow height measurements and the precipitation collected in the Pluvio$^2$ instruments.

435

440

## 6 Instrument Discussion

The experience gained from the deployment of the APS project provides insights for future precipitation measurement deployments in Antarctica and other similar environments. The following is a review of the lessons learned and conclusions from the APS field personnel based on the two years of deployment in the northwest region of the Ross Ice Shelf. The Pluvio$^2$ precipitation gauge performed very well during the deployment, other than the issue of capping within the opening. The problems with the snow capping are significant and resulted in limited to no observations once the Pluvio$^2$ became capped in early winter until there was sufficient radiative heating in the spring. A field team based out of McMurdo Station visited the Willie Field APS site in the early winter, May 2019, and found minimal signs of the impending capping of the three precipitation gauges. A creative engineering solution needs to be addressed before deploying again in the future, while remaining within the limited power availability of the remote year-round sites. The lower power requirements of the Pluvio$^2$ made it a good instrument to be installed with the low power requirements necessary for continuous year-round measurement. In total, there were four wind shields installed during the two-year deployment of the APS project. The standard configuration was the double-Alter wind shield installed with the Pluvio$^2$. The Pluvio$^2$ orifice, and the top of the wind shield, were installed at a height of approximately 3.4 m. The decision to place the Pluvio$^2$ at an elevated height worked out very well and minimized the interaction of the setup with blowing snow near the surface, and the collection of the drifting snow raising the snow height around the installation. The aluminum piping base for the double-Alter wind shield worked very well, other than the need for reinforcing the piping joints with bolts during the second field season. A modified design will be necessary for longer duration deployments to allow for a simplified raising of the Pluvio$^2$ and double-Alter wind shield due to the general increase in snow height at most locations in Antarctica. The installation of the DFAR was a failure as described in Section 3.2. During the few months in early 2018, prior to the precipitation gauge in the DFAR becoming buried, the observations indicate the precipitation gauge installed inside the DFAR was measuring the largest amount of precipitation, as would be expected with the DFAR versus a double-Alter wind shield. Unfortunately, the number of precipitation events during that time was not enough to develop the planned transfer function to estimate the corresponding accumulated precipitation in a double-Alter wind shield in comparison to the DFAR. There is still a definite need to test the existing transfer function between a double-Alter and DFAR to determine if it holds up in the Antarctic environment. However, it is anticipated that a successful installation of a DFAR would require being installed at an elevated height of approximately 4 – 5 m above the snow surface to stay above the local accumulation from the drifting snow. Such a DFAR installation would require a revised design of the DFAR, including the use of a metal base and structure, requiring a significantly higher cost and field resources for deployment. The premier site had a modified configuration during the second year because of the DFAR becoming buried, as was discussed in Section 3.2. The observations indicate the double-Alter shield provided the largest collection of precipitation followed by the single-Alter and then the unshielded precipitation gauge. The unshielded configuration collected so little precipitation it could be classified as barely usable. The bucket for the

unshielded Pluvio[2] was found to be empty when visited for site removal in November 2019. It appeared that whatever limited snow had been collected during the year was either removed by the wind or sublimated away.

The OPD-104 was a very useful instrument for the APS installation. The OPD-104 provided a low-power option, able to be powered continuously year-round, to monitor the number of particles in the air. The OPD-104 provided the ability to monitor the conditions of falling or blowing snow and act as a trigger to activate higher power consuming instruments, the CCFC and installed disdrometer, to take additional measurements. The observations from the OPD-104 have only been used in a qualitative way in the science analyses for the APS project to this point. The installation of the disdrometers, Parsivel[2] and

LPM, had mixed results. Hardware and software issues at each site during the first year resulted in no observations from the disdrometers. Observations from the disdrometers were successfully logged during the second year at three of the sites. Unfortunately, limited project resources have not allowed sufficient time to review the quality of the observations. It was realized that the LPM has a considerably higher power consumption and that provided issues in maintaining power during the polar night. The decision to install the disdrometers at a 45° angle to the horizontal, as described in Sec. 2.2, will be

revisited with future deployments. It has been inconclusive if there is sufficient advantages or disadvantages to such an installation. The CCFCs operated as well as could be expected during the two-year deployments in the Antarctic environment. There were occurrences of the camera becoming covered with rime and/or blowing snow and having an obstructed view due to the inability to heat the cameras because of limited availability of power. The WAA151 wind speed sensors provided reliable data throughout the deployment, except for a few occurrences of becoming frozen due to riming

conditions. This is a common issue with wind sensors installed in this part of Antarctica based on the experience from the University of Wisconsin automatic weather station project. The SR50AT provided reliable measurements of air temperature and snow height. The increase in height of the snow surface at some of the APS sites was more than was anticipating resulting in the installed distance of the SR50AT from the instrument to the surface being less than the instrument threshold and resulting in loss of observations during the first year. For the second year the instrument was installed a sufficiently high

height to not repeat this issue at any of the sites.

**7 Data Availability**

The data from the four APS sites can be retrieved from the United States Antarctic Program Data Center (https://www.usap-dc.org/view/dataset/601441, last accessed: 10 May 2021). The APS data can also be accessed at (Seefeldt, 2021; doi.org/10.15784/601441). All data files are comma-delimited, text files with a header line indicating the columns in the

data. The plots of APS data are png files. The photos from the CCFC camera are in jpeg format and the movies are in mp4 format. Readme pdf files are included with the data collection to provide additional information on the data availability and history.

**8 Code Availability**

All python code used for the processing, quality control, and creating of plots is available by request to the corresponding author. Additionally, the programs used in the Campbell Scientific CR6 dataloggers are available by request to the corresponding author.

**9 Conclusions**

Four APS sites were installed across the northwest Ross Ice Shelf from 2017 to 2019 providing two years of in situ measurement of precipitation in Antarctica. The APS sites were deployed to provide a direct measurement of precipitation providing a greater understanding of precipitation in Antarctica, comparison to and evaluation of atmospheric numerical models, and new insights in understanding the AIS, SMB, and changes in the GMSL in a warming climate. The instruments at each APS site included a weighing precipitation gauge installed inside a wind shield, anemometer, thermometer, sonic ranging sensor for snow height, particle counter, disdrometer, radiation sensors, and a camera. The premier site, Willie Field APS, included two additional weighing precipitation gauges and a second anemometer for experimental configurations that changed over the course of the installation. Three field seasons were completed as a part of the APS deployment covering the installation, maintenance, and removal of the APS sites. The measurements were collected in Boulder, Colorado from the combination of an internet and a radio link to the APS dataloggers. The data was processed, and quality controlled, for use in the ongoing science analyses and for sharing with the larger community. Plots of the quality-controlled data spanning the duration of data collection, monthly, and 10-day time periods have also been created for community use. The data, quality-controlled and non-quality-controlled, images and videos from the cameras, and plots have been posted to the USAP Data Center (Seefeldt, 2021; doi.org/10.15784/601441). The APS project was successful in providing in situ measurement of precipitation at remote sites in Antarctica using a low-power, autonomous measurement system. Lessons have been learned in ways to modify and improve the APSs for potential future redeployment in the Antarctic or the Arctic.

**10 Author Contributions**

MS led the APS project, contributed to the APS design, co-led the field work, led the data collection and quality control, and prepared the manuscript with contributions from all authors. TL assisted MS in the data collection, quality control, and creating of plots. SL led the APS design and fabrication and co-led the field work. TN assisted with the field work and was responsible for the GPS installation and configuration.

**11 Competing Interests**

The authors declare that they have no conflict of interest.

## 12 Acknowledgments

The authors thank the United States Antarctic Program and the Antarctic Support Contract for the organization of the field season, field assistance, and support throughout the field work based out of McMurdo Station. UNAVCO provided the power systems for the APS sites and personnel for the installation of the power systems and assisting with the APS installation, maintenance, and removal. Thanks goes to the University of Wisconsin Automatic Weather Station project, with particular recognition of Carol Constanza, for assistance with the APS installations. The authors also thank Al Jachcik and Justin Lentz of the National Center for Atmospheric Research for their work in designing and fabricating the APS equipment and datalogger programming. Thanks goes to Kristine Larson of the University of Colorado – Boulder for leading the GPS-IR data processing and interpretation of the GPS-IR results.

## 12 Financial Support

The APS project was funded by the National Science Foundation grant numbers PLR 1543377 and PLR 1543325.

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
