# Peer review of "Remote and Autonomous Measurements of Precipitation for the Northwest Ross Ice Shelf, Antarctica"

_Earth System Science Data, 2021_

## Author Response (AR1)

We thank both anonymous referees for their efforts and comments in providing the review of the manuscript. We have addressed most all of the concerns with the details included below. Explanations are provided below as well for the points which were not able to be addressed. We feel the end product is a much improved manuscript.

**RC1: Anonymous Referee #1, 10 Aug 2021**

The manuscript of Mark W. Seefeldt et al. provides an impressive overview about a multi-year precipitation dataset from 4 different Antarctic sites. From the description of the three 1-year periods the efforts, which were required to be taken in order to realize the data acquisition, become obvious.

Most points are well addressed within the manuscript. I don't see need for major revisions. The content fits well into the scope of the selected journal. During reading, it becomes also clear that the manuscript is definitely no scientific article, but a worthwhile technical dataset report.

However, I would still like to ask the authors to read and address the following minor comments. From my feeling, seeing them addressed in the manuscript would increase the information content significantly.

Minor comments:

Introduction:
- In the second sentence the authors mention big challenges (small amount of precip, distinguishing between falling snow and blowing snow) but don't provide references which discuss them (or 'introduce' them as a challenge). Could the authors provide some fitting references?
  References have been added.

Figures 2,3,4, and description of instruments in Section 2.2/2.3:
- It would be really helpful to name the single instruments (e.g., add numbers to each instrument and add a legend). For instance, I was not able to identify the location of the ODP-104. It would anyway make the instrument description section much easier. E.g., table 2 could provide the respective instrument numbers which I suggested to add to Figures 2, 3, and 4.
  Text labels have been added to all of the photos.

- Why is the Parsivel² on Figure 2 tilted (by 45°)? I contacted OTT Hydromet company and asked if there is any possibility to correct the effects of a tilted disdrometer on the derived fall velocities, particle size, precipitation rates, kinetic energy, etc (as the rotation also reduces the effective detection area of the band laser). OTT responded and told me that there's no option to modify the raw-data processing, nor was there ever any request to implement a modified data processing algorithm into the firmware. In general, I missed a description of the setup and performance of Parsivel² in Section 2.x. Would be nice to have one added. Maybe the tilting was due to another focus? Detection of blowing snow? Would be really nice to read about this!
  The following text has been added to the description of the disdrometers in Sec. 2.2:
  *The disdrometers were installed at a 45° angle to the horizontal (Fig. 2) to improve the ability to capture the horizontally moving blowing snow. The instrument algorithms for fall velocities, precipitation rates, and kinetic energy are not able to accommodate this tilt of the instrument. This was not considered to be an issue with the APS installation as the primary use of the disdrometers was to capture particle counts and sizes to better attempt to distinguish between blowing and precipitating snow.*

  The following two sentences are included in the new Sec. 6 – Instrument Discussion, regarding this topic:

*The decision to install the disdrometers at a 45° angle to the horizontal, as described in Sec. 2.2, will be revisited with future deployments. It has been inconclusive if there is sufficient advantages or disadvantages to such an installation.*

Section 3.3:
- What happened to DFAR in season 3? Did the 'unshielding' improve anything? Did it have negative effects?
  The DFAR was removed during the second field season at outlined in section 3.2. The Pluvio$^2$ precipitation gauge from the DFAR was re-installed for the second year of data collection in an unshielded configuration. I am guessing that this is what was is being inquired with these questions. The conclusions from this configuration are included in the new section on Instrument Discussion, as described below.

Snow height measurements:
- Figures 5 and 6 show sudden jumps in the measured snow height which are reported to be caused by instrument maintenance or relocation. Can the authors provide a recommendation about how to interpret the data? For sure, the snow height timelines cannot just be used to interpret snow accumulation. So what needs to be done in order to make the snow height measurements usable/interpretable? Ideally, the authors should provide a cases study of the correction procedure (to remove the instrumental effects from the actual snow accumulation evolution).
  The following two sentences have been added to the text to indicate that the Bucket NRT values and observations of precipitation are limited to event-by-event analyses.
  *The Bucket NRT values are limited to event-by-event observations of precipitation for this dataset and APS deployment. This limitation is due to the capping of the Pluvio$^2$ instrument, not factoring in evaporation (loss) occurring in the bucket, and the emptying of the bucket at the sites in November 2018.*

  There are a variety of the methods and analyses that can be used by sonic ranging sensors and these are discussed in the past studies covered in the literature review. The intent of this manuscript is to not highlight a specific method of analyses and interpretation of the data but to provide the data and any caveats with it, such as the repositioning of the sensor during the November 2018 field season.

Section 5 and Conclusions:
- I was somewhat disappointed after having read the whole manuscript. No recommendation for which instrument is the best one??? Can't the authors provide a suggestion about which kind of instrument should be preferred? I definitely see need to add a paragraph into the conclusions section in which the pros and cons of the individual instruments are discussed. Or are there other publications, which provide some guide? Some statement about recommendations could also be added to the end of Section 5, where Fig. 6d is discussed (Lines 433-436).
  Based on this comment and a similar comment by RC2, a new section: "6 Instrument Discussion" has been added to provide a review and thoughts on the performance and results from the instruments installed as a part of the APS project. There is not a recommendation as to which is the best, or which kind of instrument should be preferred, because each instrument was installed with a specific purpose that is different from the others. Recommendations on which instruments did and did not provide the most value to the project goals are included.

Typos/Grammar:
- Line 27: "..on Earth and it is …"
  Corrected

- Line 70: "…measurement studies …"

- Line 423: Suggest to modify: "Figure 6e shows the large variability in snow height during high wind…."
  Modified as suggested

**RC2: 'Comment on essd-2021-163', Anonymous Referee #2, 12 Sep 2021**

Synopsis / General comments

The paper presents a dataset of a two-year installation of autonomous precipitation measurements in Antarctica. The paper is generally well-written and explains the measurement setup and the challenges of observing precipitation in this adverse environment.

Some points are however missing (partly also addressed in the specific comments below):

- How were the quality control criteria defined?
  One of the advantages of handling only four sites and for two years of observations is that it provides the capability to manually view all of the observations multiple times. It was in this iterative process that the QC algorithms were developed subjective in producing a clean dataset when viewed manually. In the end there were four criteria to applied to all four sites and 30 criteria applied individually across the four sites (an average of 7 per site). The following text was added to provide additional information on the manual and subjective development of the automated QC processing.
  *The automated QC processing was developed through iterations of manually viewing the observation dataset spanning the two years for each APS site. Suspect and/or erroneous values were identified, and algorithms were developed to automatically remove the points in the data processing.*

  *For example, the datapoint is removed when the absolute difference of the Bucket NRT value and a rolling mean of 10 minutes of observations is larger than 10 mm LWE. Another example is the removal of wind speed observations of 0 m s$^{-1}$ for more than 180 minutes. This is an indication of the 3-cup anemometer being stuck, likely due to being frozen from riming or some other icing of the instrument. The result is an automated and objective QC processing developed using manual analyses and subjective criteria to create a clean dataset.*

- How do the disdrometer data compare to the OPD count during "events"? It would be good to get some information on that.
  At this point, there has not been the time or resources to do an analysis of comparing the disdromter data to that OPD counts. The science and analyses that have thus far been completed have focused on fulfilling the science goals of the funded project.

- How did the different Pluvio shielding perform? Which setup do you consider as best?
  See the next comment regarding the new section on Instrument Discussion.

- The conclusions are very short and do not contain much useful information. Which lessons did you learn, which instruments provided particularly useful data, and which recommendations do you have for future precipitation observations in Antarctica?
  In agreement with this comment and a comment with Referee #1, a new section has been added: "6 Instrument Discussion". These suggested questions have been addressed in that section.

Specific comments:

- Fig. 1: The choice of colors for the background is not very intuitive. I would suggest blue for water, white for ice shelf and grey for the land area.
  This figure was created by the GIS and plotting services provided by the Polar Geospatial Center at the University of Minnesota. The authors do not have the capability to modify this figure.

- Tables 1 and 2: The station is called "Willie Field", in the map it is marked as "Williams Field", please be consistent with names.
  The location of the AWS and APS observations are made near the Williams Field Skiway as a part of the United States Antarctic Program (USAP). The University of Wisconsin AWS project has referred to the site with the colloquial reference to the skiway by members of the USAP as "Willie Field". Given the co-location of APS and AWS observations the text uses the "Willie Field" reference matching that of the AWS program.

- Line 254-255: "This snow height measurement is similar to that of the snow height measurement from the sonic ranging sensor except with very different characteristics": What do you mean with that? To me this is a contradictory statement. The GPS method is definitely not similar to the ultrasonic one.
  The poorly worded statements are in an attempt to distinguish between the comparison on smaller timescales to that of larger times scales. The text has been rewritten as follows:

  *This pattern of snow height measurements by GPS-IR is similar to that of the observations from the sonic ranging sensor over the range of weeks to months. However, there are different characteristics in how the values are measured and in the pattern of observations over hours to days.*

- Line 353-355: How did you define the unique thresholds? How and where are the thresholds documented? Please comment on that!
  Please see the above reply regarding the general comment on how the quality control criteria are defined.

- Line 419-421: Is there any explanation that the two events with the highest windspeed also have the highest accumulation? Are precipitation and wind speed really correlated or does the wind bring large amounts of snow from the surroundings into the bucket? That would explain that snow height does not change much during these events.
  The purpose of this manuscript is to provide a description of the data collected from the Antarctic Precipitation System project and to direct interested users to where the data can be accessed. The description of the data includes the instruments used, installation and data collection methods and caveats, and data processing methodology. Section 5 with the Plots and Sample Data Analysis is provided to be a demonstration of a type of analysis that could be completed with the data, but it is not meant to dive into an actual scientific analysis and conclusions. Such more in-depth studies using the observations and providing an understanding of characteristics, event-by-event climatology, and forcing of different events be covered in forthcoming publications. Diving into such explanations in this manuscript is beyond the scope of this manuscript.

- Line 434-436: You show a light wind period to compare between the different Pluvio instruments. Which results does the different shielding bring at higher wind speeds? Would be good to show an example in Fig. 7 with wind speeds well above 10 m/s to get a better impression on the differences.
  Similar to the previous point, while we agree that there is more to be shown with the analyses for the different wind shield options for the Pluvio[2] instruments such an analysis is beyond the scope of this manuscript, which is already long for a data-focused manuscript.